# Invariance Properties of the Entropy Production, and the Entropic Pairing of Inertial Frames of Reference by Shear-Flow Systems

**DOI:** 10.3390/e23111515

**Published:** 2021-11-15

**Authors:** Robert K. Niven

**Affiliations:** School of Engineering and Information Technology, The University of New South Wales, Canberra, ACT 2600, Australia; r.niven@adfa.edu.au

**Keywords:** entropy production, invariance properties, Lie symmetries, inertial frames of reference, negentropy, shear flow systems

## Abstract

This study examines the invariance properties of the thermodynamic entropy production in its global (integral), local (differential), bilinear, and macroscopic formulations, including dimensional scaling, invariance to fixed displacements, rotations or reflections of the coordinates, time antisymmetry, Galilean invariance, and Lie point symmetry. The Lie invariance is shown to be the most general, encompassing the other invariances. In a shear-flow system involving fluid flow relative to a solid boundary at steady state, the Galilean invariance property is then shown to preference a unique pair of inertial frames of reference—here termed an *entropic pair*—respectively moving with the solid or the mean fluid flow. This challenges the Newtonian viewpoint that all inertial frames of reference are equivalent. Furthermore, the existence of a shear flow subsystem with an entropic pair different to that of the surrounding system, or a subsystem with one or more changing entropic pair(s), requires a source of negentropy—a power source scaled by an absolute temperature—to drive the subsystem. Through the analysis of different shear flow subsystems, we present a series of governing principles to describe their entropic pairing properties and sources of negentropy. These are unaffected by Galilean transformations, and so can be understood to “lie above” the Galilean inertial framework of Newtonian mechanics. The analyses provide a new perspective into the field of *entropic mechanics*, the study of the relative motions of objects with friction.

## 1. Introduction

In his major life’s work, Isaac Newton provided the three laws of motion that constitute what is now described as *classical* or *Newtonian mechanics* [1,2,3]:(1)First law (law of inertia): *Every body perseveres in its state of rest, or of uniform motion in a right line, unless it is compelled to change that state by forces impressed thereon*.(2)Second law (equation of motion): *The alteration of motion is ever proportional to the motive force impressed, and is made in the direction of the right line in which that force is impressed.*(3)Third law (law of action and reaction): *To every action there is always opposed an equal reaction, or the mutual actions of two bodies upon each other are always equal, and directed to contrary parts.*

The first two laws invoke the concept of an *inertial frame of reference*, defined as a frame of reference (a coordinate system) that is not undergoing acceleration, i.e., which is either at rest or in motion with a constant velocity. In contrast, a *non-inertial frame of reference* is undergoing acceleration, due to changes in velocity and/or direction (such as rotation). Newtonian mechanics builds upon the earlier viewpoint of Galileo [4], now referred to as *Galilean invariance*, that the laws of motion are the same in all inertial frames of reference. In contrast, non-inertial frames require additional correction terms (inertial forces or “fictitious forces”) to the first and second laws of mechanics, and so are not self-contained. For this reason, inertial frames of reference are privileged over non-inertial frames in Newtonian mechanics. Apart from this distinction, all inertial frames of reference are considered to be equivalent. An important corollary of this statement—unappreciated in Newton’s time, but now viewed as fundamental—is that there is no “preferred” or “absolute” inertial frame of reference for the universe.

The process of conversion between two inertial frames of reference is referred to as *Galilean transformation*. It is readily shown that the major differential and integral conservation equations of fluid mechanics, including of fluid mass, momentum and energy—as well as subsidiary equations such as the Navier–Stokes equations—remain unchanged under Galilean transformation [5], and are, therefore, Galilean invariant. They also exhibit other important invariance properties, including invariance to certain dimensionless transformations [5,6,7,8,9,10,11,12,13,14,15,16,17,18,19,20,21,22,23,24], invariance to fixed displacements in the time or space coordinates [5,20], invariance to fixed reflections or rotations of the coordinate system [5,20] and, more generally, invariance to the one-parameter Lie group of point transformations, e.g., [25,26,27,28,29,30]. Further invariance properties are also satisfied by some conservation laws: for example, the continuity equation is invariant to reversal of the time coordinate, whereas the momentum and energy equations (including the Navier–Stokes equations) are not.

The aim of this study is to examine the invariance properties of the thermodynamic entropy production, in its different formulations. This work is set out as follows. In Section 2, we provide the major equations for the entropy production, including its global (integral), local (differential), local bilinear, and global macroscopic formulations. In Section 3, we examine the invariance properties of these equations, including dimensional scaling, invariance to fixed coordinate displacements, rotations or reflections, Galilean invariance, and one-parameter Lie symmetries. Of these, the Lie invariance provides a general framework for dimensional analysis and is shown to encompass the other invariances.

In Section 4.1 and Section 4.2, we then examine the macroscopic entropy production for shear flow systems—consisting of fluid flow relative to a solid boundary—at steady state. This reveals a peculiar invariance property of such systems, in that they preference a unique pair (an *entropic pair*) of inertial frames of reference, from the infinite set of equivalent such frames for the system. In Section 4.3, we draw out an apparent paradox arising from the coexistence of different entropic pairs for different parts of the same system. In each case, the paradox reveals the presence of at least one independent source of negentropy—a power source scaled by the absolute temperature—being depleted by one or more of the different parts. Extended forms of the entropic pairing property are examined in Section 4.4 for a wide range of other steady-state shear flow systems, and in Section 5 for unsteady shear flows, presented as a series of governing principles. The conclusions of this study are then presented in Section 6.

## 2. Thermodynamic Entropy Balance and Entropy Production

Although overlooked by many fluid mechanicists, the production of entropy is fundamental to studies of nonequilibrium or irreversible processes, and therefore all flow systems in which there is friction (dissipation), diffusion, or chemical reaction. From the second law of thermodynamics, the thermodynamic entropy is not conserved; however, once created it cannot be destroyed. We, therefore, say that entropy is *preserved* [31]. From the Reynolds transport theorem for the motion of a body of fluid (the *fluid volume*, FV) through a defined region of space (the *control volume*, CV), we can extract the following integral *law of preservation* for the thermodynamic entropy [31,32,33,34,35,36,37,38,39]: (1)$$\frac{DS_{FV(t)}}{Dt}=\frac{\partial }{\partial t}\underset{CV}{∭}\rho s\;dV+\underset{CS}{∯}\rho su\cdot n\;dA=\underset{CV}{∭}\left[\frac{\partial }{\partial t}\rho s+\nabla \cdot \left(\rho su\right)\right]\;dV$$
using Cartesian spatial coordinates x [m] and time *t* [s], where *S* is the thermodynamic entropy [J K${}^{-1}$], *s* is the specific thermodynamic entropy (per unit mass of fluid) [J K${}^{-1}$ kg${}^{-1}$ = m${}^{2}$ s${}^{-2}$ K${}^{-1}$], ρ is the fluid density [kg m${}^{-3}$], u is the fluid velocity vector [m s${}^{-1}$], D/Dt is the substantial or material derivative [s${}^{-1}$], ∂/∂t is the partial time derivative [s${}^{-1}$], ∇ is the gradient operator in Cartesian coordinates [m${}^{-1}$], dV is an infinitesimal volume element [m${}^{3}$], dA is an infinitesimal area element on the boundary [m${}^{2}$], and n is the unit outward normal. The left-hand side of (Equation 1) refers to the rate of change of entropy in the fluid volume FV(t) as a function of time, while the next two integrals are calculated, respectively, over the control volume CV coincident with the fluid volume at time *t*, and its control surface CS. The second and third parts of (Equation 1) are equivalent by Gauss’ divergence theorem. Usually, the left hand side of (Equation 1) is further separated by the de Donder method into internally- and externally-driven components, respectively:(2)$$\frac{DS_{FV(t)}}{Dt}=\frac{D_{i}S_{FV(t)}}{Dt}+\frac{D_{e}S_{FV(t)}}{Dt}=\dot{\sigma }+\frac{D_{e}S_{FV(t)}}{Dt}$$
The first term in the expanded part of (Equation 2) is the internal rate of entropy production (or rate of entropy generation [40,41,42]), commonly designated $\dot{\sigma }$ [J K${}^{-1}$ s${}^{-1}$], for example, due to dissipative frictional or chemical processes. From the second law of thermodynamics, this must be nonnegative. This can be further decomposed in terms of the local rate of entropy production $\hat{\dot{\sigma }}$ [J K${}^{-1}$ m${}^{-3}$ s${}^{-1}$]:(3)$$\dot{\sigma }=\underset{CV}{∭}\hat{\dot{\sigma }}\;dV\geq 0$$
The second term in the expanded part of (Equation 2) is the externally driven rate of change of entropy, for example, due to nonfluid flows, and is commonly represented by: (4)$$\frac{D_{e}S_{FV(t)}}{Dt}=-\underset{CS}{∯}j_{S}\cdot n\;dA=-\underset{CV}{∭}\nabla \cdot j_{S}\;dV$$
where $j_{S}$ is the local nonfluid entropy flux [J K${}^{-1}$ m${}^{-2}$ s${}^{-1}$], with the negative sign arising from the sign convention $j_{S}>0$ for outward flows. Assembly of (Equation 1)–(Equation 4) and rearrangement gives an integral equation for the entropy production [31,32,33,34,35,36,37,38,39]:(5)$$\dot{\sigma }=\underset{CV}{∭}\hat{\dot{\sigma }}\;dV=\underset{CV}{∭}\left[\frac{\partial }{\partial t}\rho s+\nabla \cdot \left(j_{S}+\rho su\right)\right]\;dV\geq 0$$
Equation (Equation 5) must apply to every control volume, including each infinitesimal element dV, so from the fundamental lemma of the calculus of variations [43,44] it can be decomposed to give a differential equation for the local entropy production [31,32,33,34,35,36,37,38]:(6)$$\hat{\dot{\sigma }}=\frac{\partial }{\partial t}\rho s+\nabla \cdot \left(j_{S}+\rho su\right)\geq 0$$
The local entropy production field $\hat{\dot{\sigma }}$ must be everywhere nonnegative, otherwise it would be possible to construct control volumes for which (Equation 5) is negative.

We now consider the definition of steady-state flow. For a differential system, we can adopt the strict definition ∂(ρs)/∂t=0 for each fluid element, giving from (Equation 6):(7)$$\hat{\dot{\sigma }}=\nabla \cdot \left(j_{S}+\rho su\right)\geq 0$$
For macroscopic flow systems, however, it is more meaningful to consider the *mean steady state* [31] defined by the time average (or some other average) of (Equation 5):(8)σ˙¯=∂∂t∭CVρsdV¯+∯CS(jS+ρsu)·ndA¯=∯CS(jS¯+ρsuA¯)·ndA=∭CV∇·(jS¯+ρsuA¯)dV≥0
where $\bar{a}={t_{max}}^{-1}\int_{0}^{t_{max}}a\left(t\right)dt$ denotes the time average of some quantity *a* over an appropriate time period $t_{max}$. As evident, at the mean steady state the entropy ρsdV at any position may be fluctuating in time, but its integrated total remains constant. In either case (Equation 7) or (8), the concept of steady state precludes net rotational motion, except for small fluctuations in the time mean formulation (8).

For nonradiative processes, by substitution of the Gibbs equation and local conservation laws, the local entropy production (Equation 6) can be reduced to the bilinear form [31,32,34,35,36,37,38,39]:(9)$$\hat{\dot{\sigma }}=\sum_{k}j_{k}\cdot F_{k}$$
based on conjugate pairs of fluxes or rates $j_{k}$ and thermodynamic forces or gradients $F_{k}$, selected from those for the transport of heat, chemical species, momentum, or charge, or from the rates of chemical reaction processes. The diffusive fluxes of chemical species and charged particles in (Equation 9) are usually defined relative to the mass-average velocity u of the fluid [34,36,37], with other choices also possible [37]. In simple systems, (Equation 9) can be manifested at macroscopic scales [35,38].

Finally, various expressions have been derived for the bulk or macroscopic entropy production in different flow systems. Commonly this is expressed in terms of observable parameters, e.g., for a steady-state system [40,41,42,45]:(10)$$\bar{\dot{\sigma }}=\frac{P}{T}$$
where *P* is the power [W = J s${}^{-1}$], equal to the rate of work loss or energy dissipation, and *T* is a reference temperature [K]. Equation (Equation 10) should be equivalent to the steady-state time average (8). However, it has the advantage of avoiding the well-known “problem of closure”, due to means of products of fluctuating quantities within $\bar{\rho su}$ that cannot readily be quantified [31,40,45]. Several forms of (Equation 10) are examined further in Section 4.

## 3. Invariance Properties

We now examine several invariance properties of the entropy production, firstly in its local or differential form (Equation 6), here numbered in Roman numerals:(I)*Dimensionless invariance*: As established by nearly two centuries of mathematical and physical insight [5,6,7,8,9,10,11,12,13,14,15,16,17,18,19,20,21,22,23,24], a fundamental property of a conservation equation is its invariance to dimensionless transformations. Carvallo [10] and Vaschy [12] referred to this as *similitude*, later also described as *similarity* or *self-similarity*, e.g., [5,15,46,47,48,49,50,51,52,53] (n.b., some researchers use similarity as the general term, reserving self-similarity for systems with a solution similar to itself, such as a fractal [21]). Birkhoff [20] interpreted similarity from the perspective of group theory, as invariance under the “dimensional group of positive scalar transformations of units”. Some researchers, e.g., [18,22,23] distinguish between *complete similarity* or *self similarity of the first kind*, which can be revealed by dimensional analysis alone, and *incomplete similarity* or *self similarity of the second kind*, which cannot be revealed purely by dimensional analysis, due to divergent asymptotic behaviour of the governing equations in the limit of one or more dimensionless groups. The second category has been analyzed by the method of intermediate asymptotics [21,22,23], shown to be closely related to the method of renormalization groups, e.g., [22,23,25,54].To transform the local entropy production, we choose an appropriate length scale *L*, velocity scale $u_{0}$, temperature scale $T_{0}$, and density scale $\rho_{0}$, to define the following dimensionless variables:
(11)$$\begin{array}{c} \theta =\frac{u_{0}t}{L},\;X={\left[X,Y,Z\right]}^{⊤}=\frac{x}{L},\;={\left[U,V,W\right]}^{⊤}=\frac{u}{u_{0}},\;R=\frac{\rho }{\rho_{0}} \end{array}$$
where θ,X,U, and *R* are, respectively, the dimensionless time, position, velocity, and density. Using the scaling parameters, we also construct the following dimensionless groups:
(12)$$\begin{array}{cc} \hat{\dot{\Sigma }} & =\frac{\hat{\dot{\sigma }}LT_{0}}{\rho_{0}u_{0}^{3}}=\mathrm{dimensionless}\;\mathrm{local}\;\mathrm{entropy}\;\mathrm{production} \end{array}$$
(13)$$J_{S}=\frac{j_{S}T_{0}}{\rho_{0}u_{0}^{3}}=\mathrm{dimensionless}\;\mathrm{entropy}\;\mathrm{flux}$$
(14)$$\begin{array}{cc} \varsigma & =\frac{sT_{0}}{u_{0}^{2}}=\mathrm{dimensionless}\;\mathrm{specific}\;\mathrm{entropy} \end{array}$$The geometric scaling conditions {θ,X} impose the condition of *geometric similarity*; the velocities and fluxes $\left\{U,J_{S}\right\}$ impose the condition of *kinematic similarity*; while the remaining groups $\left\{R,\hat{\dot{\Sigma }},\varsigma \right\}$ impose the equivalence of forces, or *dynamic similarity* [47,48,49,50,51,52].Substitution of (Equation 11)–(14) into (Equation 6) and simplification gives the nondimensional entropy production equation:
(15)$$\begin{array}{c} \hat{\dot{\Sigma }}=\frac{\partial }{\partial \theta }R\varsigma +\nabla_{X}\cdot \left(J_{S}+R\varsigma U\right)\geq 0 \end{array}$$This system consists of 17 dimensional parameters $\left\{L,u_{0},\rho_{0},T_{0},t,x,u,\rho ,\hat{\dot{\sigma }},j_{S},s\right\}$, counting each vector component and the four introduced scaling parameters, with 4 classical dimensions {s, m, kg, K} (here written in SI units rather than the usual dimensional notation). Applying the Buckingham Π theorem [15], we confirm that this yields a nondimensional equation between 17−4=13 dimensionless groups $\left\{\theta ,X,U,R,\hat{\dot{\Sigma }},J_{S},\varsigma \right\}$.For anisotropic systems, (Equation 15) can be rewritten in the more comprehensive formulation:
(16)$$\begin{array}{c} \hat{\dot{\Sigma }}=\frac{\partial }{\partial \theta }R\varsigma +\nabla_{X}\cdot \left(\Psi_{S}⊙\varsigma U+R\varsigma U\right)\geq 0 \end{array}$$
which incorporates the vector dimensionless group:
(17)$$\Psi_{S}={\left[\Psi_{SX},\Psi_{SY},\Psi_{SZ}\right]}^{⊤}=j_{S}⊘\left(\rho_{0}su\right)={\left[\frac{j_{Sx}}{\rho_{0}su},\frac{j_{Sy}}{\rho_{0}sv},\frac{j_{Sz}}{\rho_{0}sw}\right]}^{⊤}$$
where ⊙ is the element-wise (Hadamard) product of two vectors to give a vector, and ⊘ is an element-wise division operator between two vectors. As evident, $\Psi_{S}$ expresses the component-wise ratios of the nonfluid and fluid entropy fluxes. Other dimensionless forms of (Equation 6) can also be derived using different choices of the reference parameters, e.g., [40,41,42].In either form (Equation 15) or (Equation 16), the nondimensional entropy production equation is invariant to transformations that maintain the same values of the dimensionless groups (Equation 11)–(14) (or with (Equation 17)), known as *dimensionless invariance*. Provided that the assumptions inherent in the derivation of (Equation 6) do not break down, such systems satisfy the properties of geometric, kinematic, and dynamic similarity, and will exhibit the same physics regardless of their absolute scale.(II)*Invariance to fixed displacements in the time or position coordinates:* For this, we adopt the modified dimensionless coordinates [5,20]:
(18)$$\begin{array}{c} \theta =\frac{u_{0}\left(t-t_{0}\right)}{L},\;X=\frac{x-x_{0}}{L} \end{array}$$
based on constant displacements $t_{0}$ and $x_{0}$, respectively, in the time and position coordinates, with the velocities, density, and other dimensionless variables unchanged. This transformation of (Equation 6) returns (Equation 15), hence (Equation 15) is invariant to fixed displacements of the time or position coordinates.(III)*Invariance to fixed reflections or rotations of the coordinate system:* Here we define a third-order coordinate rotation or reflection matrix *A*, an orthogonal matrix composed of direction cosine terms $a_{ij}$, giving the modified dimensionless positions, velocities, and fluxes [5,20]:
(19)$$\begin{array}{c} X=\frac{A^{-1}x}{L},\;=\frac{A^{-1}u}{u_{0}},\;J^{{(}_{)}}S=\frac{A^{-1}j_{S}T_{0}}{\rho_{0}u_{0}^{3}} \end{array}$$Transformation of (Equation 6) based on these coordinates, with the time, density, and other dimensionless variables unchanged, yields—with some effort—(Equation 15). The latter is therefore invariant to fixed reflections or rotations of the coordinate system.(IV)*Time antisymmetry:* Here we consider reversal of the time coordinate, and also of the velocities and fluxes, in dimensionless form [5]:
(20)$$\begin{array}{c} \theta =-\frac{u_{0}t}{L},\;U=-\frac{u}{u_{0}},\;J_{S}=-\frac{j_{S}T_{0}}{\rho_{0}u_{0}^{3}} \end{array}$$
with the positions, density, and other dimensionless variables unchanged. Transformation of (Equation 6) then yields the negative of (Equation 15) on the right-hand side. Equation (Equation 15) therefore exhibits antisymmetry with respect to time reversal, as expected for an entropy production equation.(V)*Galilean invariance:* For this we adopt the dimensionless velocities and fluxes [5,20]:
(21)$$\begin{array}{c} X=\frac{x-ct}{L},\;U=\frac{u-c}{u_{0}},\;J_{S}=\frac{\left(j_{S}-c\right)T_{0}}{\rho_{0}u_{0}^{3}} \end{array}$$
based on the constant velocity c, which provides a Galilean transformation between two inertial frames of reference. The time, density, and other dimensionless variables are unchanged. This transformation of (Equation 6) returns (Equation 15), hence (Equation 15) is Galilean invariant.(VI)*Lie invariance*: The invariance properties or symmetries associated with infinitesimal Lie transformations of a differential or integral equation constitutes a large topic, e.g., [25,26,55,56,57,58]. For the present study, we restrict the discussion to the one-parameter Lie group of point scaling transformations, e.g., [25,26,27,28,29,30]. For the local entropy production (Equation 6), this is defined by the 13-parameter map:
(22)$$\begin{array}{c} \left[t,x,y,z,u,v,w,\rho ,s,j_{sx},j_{sy},j_{sz},\hat{\dot{\sigma }}\right]\mapsto \\ \;\;\left[\epsilon^{\alpha_{t}}\theta ,\epsilon^{\alpha_{x}}X,\epsilon^{\alpha_{y}}Y,\epsilon^{\alpha_{z}}Z,\epsilon^{\alpha_{u}}U,\epsilon^{\alpha_{v}}V,\epsilon^{\alpha_{w}}W,\epsilon^{\alpha_{\rho }}R,\epsilon^{\alpha_{s}}\varsigma ,\epsilon^{\alpha_{j_{sx}}}J_{sx},\epsilon^{\alpha_{j_{sy}}}J_{sy},\epsilon^{\alpha_{j_{sz}}}J_{sz},\epsilon^{\alpha_{\hat{\dot{\sigma }}}}\hat{\dot{\Sigma }}\right] \end{array}$$
where ϵ∈R is a scaling parameter, $\alpha_{i}$ are the scaling exponents, and the capital or Greek letters denote the transformed variables (here, not necessarily dimensionless). Substitution into (Equation 6) and simplification gives the mathematical form (Equation 15) subject to 7 auxiliary relations for $\alpha_{\hat{\dot{\sigma }}}$, which can be solved to give:
(23)$$\begin{array}{c} \begin{array}{cccccc} \alpha_{u} & =-\alpha_{t}+\alpha_{x},\; & \alpha_{j_{sx}} & =-\alpha_{t}+\alpha_{\rho }+\alpha_{s}+\alpha_{x},\; & \alpha_{\hat{\dot{\sigma }}} & =-\alpha_{t}+\alpha_{\rho }+\alpha_{s}\\ \alpha_{v} & =-\alpha_{t}+\alpha_{y},\; & \alpha_{j_{sy}} & =-\alpha_{t}+\alpha_{\rho }+\alpha_{s}+\alpha_{y},\\ \alpha_{w} & =-\alpha_{t}+\alpha_{z},\; & \alpha_{j_{sz}} & =-\alpha_{t}+\alpha_{\rho }+\alpha_{s}+\alpha_{z} \end{array} \end{array}$$This result can be interpreted in several ways:(a)If the transformed parameters in (Equation 22) have the same dimensions as the original parameters—the standard interpretation—they must be considered as rescaled dimensional variables, while the $\epsilon^{\alpha_{i}}$ terms are dimensionless conversion factors. Transformation gives a rescaled form of (Equation 6) rather than a nondimensional equation, while the auxiliary relations (Equation 23) provide the relations between conversion factors for the dependent and independent variables. This interpretation is useful, representing a rescaling between a model and a prototype to maintain similarity, e.g., [27,28,29], or rescaling by a change of units. However, since this interpretation can be handled by the more general apparatus of dimensionless invariance (I), it need not be considered further.(b)Alternatively, if the transformed variables in (Equation 22) are intended to be dimensionless, the $\epsilon^{\alpha_{i}}$ terms must be interpreted as dimensional scaling parameters. Assuming a positive dimensionless parameter ϵ, each $\alpha_{i}$ term will carry the base-ϵ logarithm of the dimensions of quantity *i*. As an example, the relation $\alpha_{u}=-\alpha_{t}+\alpha_{x}$ carries the dimensions $\log_{\epsilon }\left(m\;s^{-1}\right)=-\log_{\epsilon }s+\log_{\epsilon }m$, written in SI units. In consequence, the auxiliary relations (Equation 23) simply express the relations between the dependent and independent dimensions within the system, enabling the transformation of (Equation 6) into the nondimensional form (Equation 15).In this example, there are 7 auxiliary relations (Equation 23) composed of 13 $\alpha_{i}$ terms, so there must be 13−7=6 independent $\alpha_{i}$ terms, hence, 6 independent dimensions of the system. We have chosen to express (Equation 6) in terms of the dimensions {s, m${}_{x}$, m${}_{y}$, m${}_{z}$, kg m${}_{x}$^−1^ m${}_{y}$^−1^ m${}_{z}$^−1^, J K^−1^ kg^−1^} of the 6 linearly independent parameters {t,x,ρ,s}, counting each vector component of x, to span the set of 6 fundamental units {s, m${}_{x}$, m${}_{y}$, m${}_{z}$, kg, K}. This choice of dimensions—which distinguishes the length scale in each Cartesian direction, and which adopts the dimensions of the fluid density and specific entropy rather than mass and temperature—is rather unexpected, but it does represent the intrinsic or “indigenous” dimensions of the local entropy production (Equation 6). Other choices of independent variables are possible, provided that they span the set of 6 fundamental units. The resulting Lie transformation (Equation 22) and (Equation 23) is consistent with the Buckingham Π theorem [15], reducing a system of 13+6=19 parameters (including the 6 independent $\alpha_{i}$ terms) and 6 dimensions to the nondimensional form (Equation 15) with 19−6=13 dimensionless groups. It is curious that the number and character of the dimensions and the vector formulation of (Equation 15) are revealed automatically by the Lie apparatus, seemingly for free!(c)We can also consider hydrid interpretations, in which the transformed parameters and the $\epsilon^{\alpha_{i}}$ terms in (Equation 22) both carry dimensions, in combination giving those of the original parameters. Such representations may be feasible, but owing to their complexity, we exclude them from further discussion.We see that the second interpretation (b) of the Lie transformation provides an alternative formulation of the dimensionless invariance (I), expressed in terms of fundamental quantities that better reflect the underlying symmetries of (Equation 6). This equivalence between dimensionless invariance and Lie point symmetry has been recognized by some researchers, e.g., [56], but is not well-known in either the mathematical or engineering literature. By remapping the scaling parameters in the manner of (Equation 18)–(Equation 21), we can also extract the other invariances (II)–(V), hence, the Lie invariance can be considered to subsume all of these invariances.In their analyses of other conservation laws, some authors deduce further equivalences, which for the present study would yield $\alpha_{x}=\alpha_{y}=\alpha_{z}$, $\alpha_{u}=\alpha_{v}=\alpha_{w}$ and $\alpha_{j_{sx}}=\alpha_{j_{sy}}=\alpha_{j_{sz}}$, e.g., [27,29]. These relations are not provided by the above Lie transformation of (Equation 6), nor are they evident from its vector structure. They can, however, be interpreted as consequences of the invariance to fixed rotations or reflections of the coordinate system (III).We note that Lie and later workers developed comprehensive algorithmic methods for the analysis of infinitesimal Lie symmetries of differential equations [20,26,28,55,57,58]. Furthermore, several researchers have suggested multiparametric extensions of the above Lie apparatus, applicable to all conservation equations [26,58,59,60]. The current level of understanding of the Lie invariances of the local entropy production (Equation 6) is therefore incomplete, and warrants further examination.

Now examining the bilinear form of the local entropy production (Equation 9), it is evident that each flux and thermodynamic force can be converted to a nondimensional form, hence, this equation also satisfies dimensionless invariance (I). Since the fluid, momentum, and heat fluxes are expressed relative to a common inertial frame of reference, while the diffusive fluxes are expressed relative to the mass-average velocity u, (Equation 9) will also be Galilean invariant (V) to a change in the common inertial frame of reference [38]. It is also readily verified that (Equation 9) satisfies equivalent forms of the other invariance principles (II)–(IV) and (VI), which it must, since it is equivalent to the differential equation (Equation 6).

Examining the integral entropy production (Equation 5), this can be decomposed into a field of constituent differential equations for $\hat{\dot{\sigma }}$, each of which satisfies the invariance properties (I)–(VI). The integral equation will therefore satisfy the same transformations, provided they are applied identically throughout the fluid and control volumes. Furthermore, the invariance to fixed displacements (II)—which does not affect any velocities, accelerations, fluxes, or rates of change—can alternatively be applied in a smoothly-varying fashion to each infinitesimal element, to select a single set of coordinates for the entire control volume. Invariance to fixed rotations or reflections (III) and Galilean invariance (V) can then be demonstrated for this global coordinate system by mathematical induction.

Finally, the macroscopic entropy production (Equation 10), being equivalent to the time average of the integral Equation (8), should satisfy the same invariance properties (I)–(VI), based on macroscopic analogs of the local dimensionless groups (Equation 11)–(14). We examine several such nondimensional transformations (I) in the remainder of this study, as well as the Galilean invariance (V) of the macroscopic entropy production. These are used to draw out an additional, previously unrecognized, invariance property of shear flow systems.

## 4. Macroscopic Shear Flow Systems and an Entropic Invariance Principle

### 4.1. External and Internal Shear Flow Systems

We now examine two macroscopic shear flow systems involving fluid flow relative to a solid boundary at steady state, to reveal a rather different invariance property of these systems. In keeping with the convention for idealized flows in fluid mechanics, in this section we consider only drag forces, and neglect lift forces, gravitational or buoyancy forces, electromagnetic fields, rotational or vibrational motions, as well as the effects of special or general relativity. We assume that solid objects are rigid and remain at constant elevation or position. While we recognize that all dissipative processes generate heat, we restrict the analysis to the relative motions of solids and fluids under (approximately) isothermal conditions, and do not consider convective flows due to temperature gradients. We return to the consideration of lift forces in Section 4.3.2(c), and body forces and accelerations in Section 5.

#### 4.1.1. External Flows

In the first example, consider the steady irrotational non-vibrational motion of a fluid relative to a solid object, such as a moving sphere or aircraft, in fluid mechanics referred to as an *external flow*, e.g., [47,48,50,51,52]. This is represented by the control volume illustrated in Figure 1a. We note that the selected inertial frame of reference, in which the fluid is considered to be moving around a stationary object, represents only one choice from an infinite set of equivalent inertial frames of reference for this system. An alternative inertial frame of reference is shown in Figure 1b, in which the ambient fluid (referenced at infinite distance) is considered stationary (in the mean), while the object is moving. Insofar as the subsystem consisting of the object and its surrounding fluid is concerned, there is no distinction in Newtonian mechanics between the flows described in these two inertial frames of reference, nor between these and any other inertial frames.

The flow regime in Figure 1a,b is commonly described by the Reynolds number [46,47,48,49,50,51,52,53,62]:(24)$$\begin{array}{c} Re_{s}=\frac{\rho d_{s}U}{\mu } \end{array}$$
where $d_{s}$ is a consistent length scale [m], usually obtained from the solid object, *U* is the mean velocity of the free stream of fluid relative to the solid [m s${}^{-1}$], and μ is the dynamic viscosity [Pa s]. We allow $Re_{s}<0$ when U<0, expressed in the coordinate system of Figure 1a. For simple geometries, the frictional and pressure drag force is expressed by the drag coefficient [46,47,48,49,50,51,52,53,62]:(25)$$\begin{array}{c} C_{D}=\frac{F_{D}}{\frac{1}{2}\rho A_{s}U^{2}} \end{array}$$
where $F_{D}$ is the (scalar) drag force of the fluid on the object [N] and $A_{s}$ is the cross-sectional area of the solid normal to the flow [m${}^{2}$]. We allow $C_{D}<0$ when $F_{D}<0$. Correlations for $C_{D}$ as a function of $Re_{s}$ are available for a range of solid shapes, typically in graphical form, e.g., [20,46,47,48,49,50,51,52,63]. For more complicated geometries, it may be necessary to adopt different drag and lift coefficients in different directions (see Section 4.3.2) or to integrate the pressure and viscous stress around the object [47,52]. Assuming subsonic flow, an absence of lift, and isothermal conditions, the macroscopic entropy production (Equation 10) is given by [40,41,42]:(26)$$\begin{array}{c} {\bar{\dot{\sigma }}}_{ext}=\frac{F_{D}U}{T}=\frac{\frac{1}{2}\rho A_{s}C_{D}U^{3}}{T}\geq 0 \end{array}$$
From (Equation 24)–(Equation 25), this can be expressed in the nondimensional form:(27)$${\bar{\dot{\Sigma }}}_{ext}=\frac{{\bar{\dot{\sigma }}}_{ext}T}{A_{s}\mu g}=\frac{F_{D}}{\rho gd_{s}A_{s}}Re_{s}=\frac{1}{2}C_{D}\frac{{Re_{s}}^{3}}{Ga_{s}}\geq 0$$
where *g* is the acceleration due to gravity [m s${}^{-2}$] and $Ga_{s}=\rho^{2}g{d_{s}}^{3}/\mu^{2}$ is the Galileo number, which expresses the ratio of gravity to viscous forces [64,65,66]. We note that in all instances, *U* and $F_{D}$ will have the same sign ([24]; see later discussions), hence, $Re_{s}$ and $C_{D}$ will also, enforcing the nonnegativity of (Equation 26) and (Equation 27) in the event of flow reversal.

#### 4.1.2. Internal Flows

In the second example, consider the steady irrotational motion of a fluid within a solid conduit such as a pipe—commonly termed an *internal flow*—under a pressure gradient (Poiseuille flow), as shown in Figure 2a. Again this choice of inertial frame of reference, in which the fluid is considered to be moving while the solid is stationary, represents only one choice from an infinite set of equivalent inertial frames of reference for this system. An alternative inertial frame of reference is represented in Figure 2b, in which the fluid is considered stationary (in the mean), while the enclosing solid is in motion. Once again there is no distinction in Newtonian mechanics between the flows described in these two inertial frames of reference, nor in any other inertial frame.

The flow regime for internal flow through a conduit of noncircular cross-section is generally described by the Reynolds number [46,50,62]:(28)$$\begin{array}{c} Re_{H}=\frac{\rho d_{H}U}{\mu }=\frac{\rho d_{H}Q}{\mu A} \end{array}$$
where $d_{H}=4A/W$ is the hydraulic diameter [m], *A* is the flow cross-sectional area [m${}^{2}$], *W* is the wetted perimeter [m], U=Q/A is now the mean cross-sectional velocity of the fluid relative to the solid [m s${}^{-1}$], and *Q* is the volumetric flow rate [m${}^{3}$ s${}^{-1}$]. The total pressure loss $p_{L}$ [Pa] can be expressed as [46,47,48,49,50,51,52,62]:(29)$$\begin{array}{c} p_{L}=\rho gH_{L}=\left(\frac{fL}{d_{H}}+\sum K_{L}\right)\frac{\rho U^{2}}{2}=\left(\frac{fL}{d_{H}}+\sum K_{L}\right)\frac{\rho Q^{2}}{2A^{2}} \end{array}$$
where $H_{L}$ is the total head loss [m], *f* is the Darcy-Weisbach friction factor [-], *L* is the flow length [m], and $K_{L}$ is the loss coefficient for a pipe fitting [-], summed over the total number of fittings. To enable flow reversal, we impose $p_{L}<0$, $H_{L}<0$, f<0 and $K_{L}<0$ (a pressure, head or frictional gain) for flows with U<0, Q<0 and $Re_{H}<0$ c.f., [66,67,68,69]. Equations (Equation 28) and (Equation 29) reduce to the expressions for a circular pipe of diameter $d_{H}=d$[46,47,48,49,50,51,52,62], while variants are available for conduits of different cross-section, e.g., [37,46,48,50,62,65,70,71]. Furthermore, *f* is a function of $Re_{H}$, with an analytical solution for laminar flow, and various correlations for turbulent flow as a function of Reynolds number and surface roughness [46,47,48,49,50,51,52,72]. For isothermal flow, the macroscopic entropy production (Equation 10) is then given by [40,41,42,66]:(30)$$\begin{array}{c} {\bar{\dot{\sigma }}}_{int}=\frac{p_{L}Q}{T}=\frac{\rho gH_{L}Q}{T}=\left(\frac{fL}{d_{H}}+\sum K_{L}\right)\frac{\rho AU^{3}}{2T}\geq 0 \end{array}$$
From (Equation 28) and (Equation 29), this can be expressed in the nondimensional form, c.f., [66]:(31)$${\bar{\dot{\Sigma }}}_{int}=\frac{{\bar{\dot{\sigma }}}_{int}T}{d_{H}L\mu g}=\frac{H_{L}}{L}\;\frac{A}{d_{H}^{2}}\;Re_{H}=\frac{1}{2}\;\left(f+\sum K_{L}\frac{d_{H}}{L}\right)\;\frac{A}{d_{H}^{2}}\;\frac{{Re_{H}}^{3}}{Ga_{H}}\geq 0$$
where $Ga_{H}=\rho^{2}g{d_{H}}^{3}/\mu^{2}$ is the hydraulic Galileo number. Since $p_{L}$ and *U* (or $H_{L}$ and $Re_{H}$) have the same sign [24], (Equation 30) and (Equation 31) will remain nonnegative in the event of flow reversal.

The dimensionless entropy production terms for external and internal flows, (Equation 27) and (Equation 31), cannot be compared directly, being nondimensionalized by different parameters. However, it is always possible to compare their dimensional values, measured in a consistent set of units:(32)$${\bar{\dot{\sigma }}}_{ext}=\frac{A_{s}\mu g}{T}\;{\bar{\dot{\Sigma }}}_{ext},\;{\bar{\dot{\sigma }}}_{int}=\frac{d_{H}L\mu g}{T}\;{\bar{\dot{\Sigma }}}_{int}$$

The above principles extend naturally to other shear flow systems, a number of examples of which are examined further in Section 4.4 and Section 5.

### 4.2. An Invariance Principle of Entropic Pairing

From the analyses in Section 3, in each of the above examples, the macroscopic entropy production (Equation 26) or (Equation 30) (or its nondimensional form (Equation 27) or (Equation 31)) must be Galilean invariant, since it arises from the time average of the integral formulation (Equation 5). Scrutinizing the examples carefully, it is evident that the velocity *U* used to define the Reynolds number (Equation 24) or (Equation 28) is not situated within a single inertial frame of reference. Instead, it represents the difference in velocity between two inertial frames of reference: (i) the frame moving with the solid or, in other words, that in which the solid is stationary, and (ii) the frame moving with the mean fluid flow, or that in which the mean flow is stationary. These correspond to the pair of frames respectively illustrated in Figure 1a,b or Figure 2a,b. In other words, the Reynolds number provides a dimensionless Galilean transformation between the two unique inertial frames of reference that define the shear flow. It is precisely for this reason that the Reynolds number—and, consequently, any derived quantity such as the entropy production—is independent of any individual inertial frame of reference, and so is the Galilean invariant.

We can take this argument further to consider that in each example, the Reynolds number (Equation 24) or (Equation 28) *preferentially selects* the solid and free-stream inertial frames of reference from the infinite set of equivalent inertial frames of reference for that system. This insight is profound, since it challenges the usual Galilean viewpoint that all inertial frames are equivalent: clearly, from the perspective of a shear flow subsystem defined by fluid flow relative to a solid boundary, they are not. Indeed, the very notion of dimensionless invariance (I) of a macroscopic shear flow system requires the existence of a special pair of inertial frames of reference, since otherwise the dimensionless groups used to represent dynamic similarity (such as the Reynolds number) could not be defined.

The two inertial frames identified by each Reynolds number could be described as a *Reynolds pair* of inertial frames of reference. However, they are not preferenced by the Reynolds number alone, but also by the drag coefficient (Equation 25), the friction factor (Equation 29), the entropy production (Equation 26) or (Equation 30), and every dimensionless group that invokes inertial processes. Examining the Reynolds number, we know that it discriminates between two flow regimes characterized by different rates of entropy production [66,73,74,75]:(a)*Laminar flow*, in which the diffusion of momentum is dominated by viscous processes, leading to a low Reynolds number and lower entropy production, and(b)*Turbulent flow*, in which the diffusion of momentum is dominated by inertial processes, leading to a higher Reynolds number and a commensurately higher rate of entropy production.
In other words, the Reynolds number is an entropic parameter, which expresses the relative influence of viscous and inertial momentum transport processes on the entropy production. For this reason, the two selected inertial frames of reference should be described as an *entropic pair* of inertial frames of reference, which are *entropically paired* by the shear flow system. We also refer to the distance between the two selected frames as their *inertial separation*, in each case given in dimensional form by the inertial velocity *U*.

Variants of the above arguments apply to all other shear flows, modified if necessary to encompass the relative motions of more than one solid or fluid. A variety of other such systems are examined in Section 4.4. In each shear flow, the entropic pair of inertial frames of reference will be invariant to the transformations (I)–(VI) identified in Section 3, and so can be identified as an additional invariance property of the system. In this respect, the invariant property of entropic pairing can be considered to “sit above” the Galilean inertial framework of Newtonian mechanics.

### 4.3. An Entropic Paradox and Its Resolution

#### 4.3.1. Statement of the Paradox

Now consider the macroscopic flow system illustrated in Figure 3, containing multiple examples of external flow subsystems (subsidiary control volumes), each involving steady irrotational non-vibrational motion between a solid object and a fluid. Such a flow system can be seen, for example, in the movements of multiple fish in the ocean, or of multiple aircraft above most inhabited parts of the Earth. As drawn, Figure 3 adopts an inertial frame of reference relative to a reference solid such as the Earth’s surface, with the ambient fluid moving at constant velocity *U* relative to this surface (ignoring boundary-layer effects). As noted, however, this inertial frame of reference is not unique and is adopted here simply for convenience. Within the fluid, we consider four shear flow subsystems created by various solid objects:(a)Subsystem A, containing Object A, which is stationary with respect to the reference solid ($U_{A}=0$), and experiences an incoming flow field of ambient velocity *U*;(b)Subsystem B, containing Object B moving at the speed $U_{B}<0$ relative to the reference solid (i.e., in motion upstream);(c)Subsystem C, containing Object C moving at the speed $U_{C}>0$ relative to the reference solid (i.e., in motion downstream); and(d)Subsystem D, consisting of an internal flow field of ambient velocity $U_{D}$ established within a container or region, within which Object D is held stationary with respect to the Subsystem D boundary and the reference solid.

Considering the ambient flow in Figure 3 to be an internal flow driven by a pressure gradient (Poiseuille flow) without fittings, its Reynolds number (Equation 28) and head loss (Equation 29) are:(33)$$\begin{array}{c} Re_{f}=\frac{\rho D_{f}U}{\mu },\;H_{L,f}=\frac{f_{f}L_{f}}{D_{f}}\frac{U^{2}}{2g} \end{array}$$
where subscript *f* denotes the ambient fluid. The dimensionless entropy production (Equation 31) for the ambient flow is therefore:(34)$${\bar{\dot{\Sigma }}}_{f}=\frac{{\bar{\dot{\sigma }}}_{f}T}{D_{f}L_{f}\mu g}=\frac{H_{L,f}}{L_{f}}\;\frac{A_{f}}{D_{f}^{2}}\;Re_{f}=\frac{1}{2}\;f_{f}\;\frac{A_{f}}{D_{f}^{2}}\;\frac{{Re_{f}}^{3}}{Ga_{f}}\geq 0$$
For the external flows associated with Objects A to D, the Reynolds numbers (Equation 24) and drag coefficients (Equation 25) are, respectively:(35)$$\begin{array}{c} \begin{array}{cccc} Re_{A} & =\frac{\rho d_{A}U}{\mu },\; & C_{D_{A}} & =\frac{F_{A}}{\frac{1}{2}\rho A_{A}U^{2}},\\ Re_{B} & =\frac{\rho d_{B}\left(U-U_{B}\right)}{\mu },\; & C_{D_{B}} & =\frac{F_{B}}{\frac{1}{2}\rho A_{B}{\left(U-U_{B}\right)}^{2}},\\ Re_{C} & =\frac{\rho d_{C}\left(U-U_{C}\right)}{\mu },\; & C_{D_{C}} & =\frac{F_{C}}{\frac{1}{2}\rho A_{C}{\left(U-U_{C}\right)}^{2}},\\ Re_{D} & =\frac{\rho d_{D}U_{D}}{\mu },\; & C_{D_{D}} & =\frac{F_{D}}{\frac{1}{2}\rho A_{D}{U_{D}}^{2}}, \end{array} \end{array}$$
based on appropriate choices of length scales, drag forces, and cross-sectional areas. In consequence, their macroscopic dimensionless entropy production terms (Equation 27) are, respectively:(36)$${\bar{\dot{\Sigma }}}_{A}=\frac{1}{2}C_{D_{A}}\frac{{Re_{A}}^{3}}{Ga_{A}},\;{\bar{\dot{\Sigma }}}_{B}=\frac{1}{2}C_{D_{B}}\frac{{Re_{B}}^{3}}{Ga_{B}},\;{\bar{\dot{\Sigma }}}_{C}=\frac{1}{2}C_{D_{C}}\frac{{Re_{C}}^{3}}{Ga_{C}}\;{\bar{\dot{\Sigma }}}_{D}=\frac{1}{2}C_{D_{D}}\frac{{Re_{D}}^{3}}{Ga_{D}}$$
where $Ga_{i}=\rho^{2}gd_{i}^{3}/\mu^{2}$ for i∈{A,B,C,D}. We can also analyze the flow in Subsystem D as a separate internal flow for which the Reynolds number (Equation 28), friction factor (Equation 29), and dimensionless entropy production (Equation 31) are given, respectively, by:(37)$$\begin{array}{c} Re_{\vec{D}}=\frac{\rho d_{\vec{D}}U_{D}}{\mu },\;f_{\vec{D}}=\frac{H_{L,\vec{D}}}{L_{\vec{D}}}\frac{2gd_{\vec{D}}}{{U_{D}}^{2}},\;{\bar{\dot{\Sigma }}}_{\vec{D}}=\frac{1}{2}\;f_{\vec{D}}\;\frac{A_{\vec{D}}}{d_{\vec{D}}^{2}}\;\frac{{Re_{\vec{D}}}^{3}}{Ga_{\vec{D}}} \end{array}$$
where $\vec{D}$ denotes the enclosed Subsystem D, $d_{\vec{D}}$ is the hydraulic diameter [m], $A_{\vec{D}}$ is the cross-sectional area [m${}^{2}$], $H_{L,\vec{D}}$ is the head loss [m], $L_{\vec{D}}$ is the flow length [m], and $Ga_{\vec{D}}=\rho^{2}gd_{\vec{D}}^{3}/\mu^{2}$. Subsystem D provides an example of an external flow subsystem nested inside an internal flow subsystem, in turn nested inside the bulk flow system. For even greater generality, we could imagine Subsystem D to be detached from the reference solid and moving at a constant velocity through the bulk fluid, requiring an additional Reynolds number, drag coefficient, and entropy production term associated with its motion.

As evident, in this example there are six different Reynolds numbers, drag or friction coefficients, and rates of entropy production, associated with the bulk flow of the ambient fluid, with each external flow in Subsystems A to D, and with the internal flow field in Subsystem D. From Section 4.2, we know that each system or subsystem has its own entropic pair of inertial frames of reference moving with its solid and its mean fluid flow. In consequence, different parts of a connected flow system can coexist with different entropically paired inertial frames of reference. This creates an *entropic paradox*: how can a shear flow system preferentially select many different—and unrelated—entropic pairs of inertial frames of reference?

#### 4.3.2. Resolution of the Paradox

To resolve this paradox, we first revisit the concept of negative entropy or *negentropy*, conceived by many prominent researchers [76,77] and named by Brillouin [78]. In this perspective, the universe contains a finite store or reservoir of negentropy, which is continually and irreversibly depleted by dissipative processes. The negentropy of the universe therefore provides a thermodynamic potential *N*, which decreases in the direction of spontaneous change. It therefore generalizes the availability, available work, free energy, affinity, and Planck potential concepts [79,80,81,82,83,84,85] and subsumes related ideas such exergy and essergy [86,87]. In a flow system, the rate of change of negentropy due to processes within a fluid volume is, by definition:(38)$$\frac{DN_{FV(t)}}{Dt}=-\frac{DS_{FV(t)}}{Dt}$$
hence, from the de Donder separation (Equation 2):(39)$$\frac{DN_{FV(t)}}{Dt}=\frac{D_{i}N_{FV(t)}}{Dt}+\frac{D_{e}N_{FV(t)}}{Dt}=\dot{\varphi }+\frac{D_{e}N_{FV(t)}}{Dt}$$
where $\dot{\varphi }=D_{i}N_{FV(t)}/Dt$ is the internally driven rate of change of negentropy or, more simply, the rate of negentropy production, and $D_{e}N_{FV(t)}/Dt$ is the externally driven rate of change of negentropy. We see that the rate of negentropy production is identical to the rate of entropy production but of opposite sign $\dot{\varphi }=-\dot{\sigma }\leq 0$, or in time-averaged dimensionless form, $\bar{\dot{\Phi }}=-\bar{\dot{\Sigma }}\leq 0$. We can also refer to the rate of negentropy consumption, defined by the absolute value $|\dot{\varphi }|=\dot{\sigma }\geq 0$ or $|\bar{\dot{\Phi }}|=\bar{\dot{\Sigma }}\geq 0$.

Now consider each subsystem shown in Figure 3 in turn:(a)*Subsystem A*: From Figure 3, we see that Object A shares the same inertial frame of reference as the reference solid (it could even be joined to it by some physical framework or magnetic coupling). The entropic pair for Subsystem A is, therefore, the same as for the bulk flow, with the inertial separation *U*. Furthermore, the entropy production ${\bar{\dot{\Sigma }}}_{A}$ (Equation 36) reveals the existence of a source of negentropy for Subsystem A, which is being continuously depleted by the reaction to the drag force, i.e., by the need to continuously do work against the frictional and pressure drag between the fluid and Object A. This rate of work is, at minimum, $\dot{W}=T_{A}{\bar{\dot{\sigma }}}_{A}$ (assuming 100% efficiency), and the associated rate of negentropy consumption is, at minimum, $|{\bar{\dot{\varphi }}}_{A}|={\bar{\dot{\sigma }}}_{A}$, or in dimensionless form $|{\bar{\dot{\Phi }}}_{A}|={\bar{\dot{\Sigma }}}_{A}$. From the information provided in Figure 3, we do not know if this source of negentropy is situated at Object A itself, for example, a source of motive power attached to the object, or if it is incurred by the source of negentropy driving the ambient fluid stream, enabling it to perform the required added work against Object A held in a fixed position. We do know, however, that there must be a source of negentropy for Subsystem A, and that this must be situated with either (or both) Object A or the ambient flow field.(b)*Subsystem B*: Now consider Subsystem B, which preferences the entropic pair of inertial frames of reference defined by the ambient flow and Object B, with the inertial separation $U-U_{B}$. In all cases for which $U_{B}<0$ (moving upstream), this entropic pair will differ from that for the ambient flow, due to the different solid velocities. There must be a source of negentropy for the entropy production ${\bar{\dot{\Sigma }}}_{B}$ incurred by Subsystem B. Furthermore—and this is the crucial point—even if there is a physical connection or coupling between Object B and the reference solid (e.g., a chassis and a set of wheels), we know that the ambient flow field can only contribute to this entropy production to the extent of its inertial separation *U* from the reference solid, i.e., for which:
(40)$$Re_{B^{\prime }}=\frac{\rho d_{B}U}{\mu },\;C_{D_{B^{\prime }}}=\frac{F_{B}}{\frac{1}{2}\rho A_{B}U^{2}},\;{\bar{\dot{\Sigma }}}_{B^{\prime }}=\frac{1}{2}C_{D_{B^{\prime }}}\frac{{Re_{B^{\prime }}}^{3}}{Ga_{B}}$$(assuming a constant length scale and cross-sectional area). Any positive excess ${\bar{\dot{\Sigma }}}_{B}-{\bar{\dot{\Sigma }}}_{B^{\prime }}$ must therefore be incurred by an independent source of negentropy associated with Object B, for example a source of motive power attached to this object, or connected to it by other means such as a magnetic coupling. Alternatively, if there is no connection between Object B and the reference solid, nor with any other solid in the system, then all of the negentropy consumption ${\bar{\dot{\Sigma }}}_{B}$ for Subsystem B, not just the excess ${\bar{\dot{\Sigma }}}_{B}-{\bar{\dot{\Sigma }}}_{B^{\prime }}$, must be incurred by the source attached to Object B.(c)*Subsystem C*: Next, consider Subsystem C, which preferences the entropic pair of inertial frames defined by the ambient stream and Object C, with the inertial separation $U-U_{C}$. The Reynolds number, drag coefficient, and entropy production are given in (Equation 35) and (Equation 36). For the downstream motion ($U_{C}>0$) of Object C, there are three scenarios:(i)For $0<U_{C}<U$, Object C will move more slowly than the downstream flow, and the subsystem will have a positive $Re_{C}$, $C_{D_{C}}$, and entropy production.(ii)For the special case $U_{C}=U$, Object C will move with the fluid stream, hence, $Re_{C}=0$, $C_{D_{C}}=0$, so (for this ideal case) there is no entropic pair of inertial frames of reference and no entropy production.(iii)For $U_{C}>U$, Object C will move more rapidly than the downstream flow, incurring a drag force in the opposite direction, again leading to a positive $Re_{C}$, $C_{D_{C}}$, and entropy production.As for Subsystem B, the ambient flow can only contribute to the entropy production of Subsystem C to the extent of its inertial separation *U*, i.e., for which:
(41)$$Re_{C^{\prime }}=\frac{\rho d_{C}U}{\mu },\;C_{D_{C^{\prime }}}=\frac{F_{C}}{\frac{1}{2}\rho A_{C}U^{2}},\;{\bar{\dot{\Sigma }}}_{C^{\prime }}=\frac{1}{2}C_{D_{C^{\prime }}}\frac{{Re_{C^{\prime }}}^{3}}{Ga_{C}}$$The excess ${\bar{\dot{\Sigma }}}_{C}-{\bar{\dot{\Sigma }}}_{C^{\prime }}$ will be negative (non-physical), zero, or positive, respectively, for the above three cases. For case (i), it is thus possible for all of the negentropy consumption ${\bar{\dot{\Sigma }}}_{C}$ to be harnessed from the ambient flow, with the solid object partly carried by the fluid. For case (ii), both terms in the excess vanish, and there is no entropy production. For case (iii), however, the positive excess reveals the existence of a source of negentropy for Object C, independent of that for the ambient flow. Alternatively, regardless of the above categories, if the negentropy harnessed by Object C from the flow is less than ${\bar{\dot{\Sigma }}}_{C^{\prime }}$, then this difference must also be provided by the independent source of negentropy for Object C.If the object is in translational motion in a different direction to the surrounding flow field, the above arguments must be modified to account for the relative velocity vectors. For example, consider an ambient flow of velocity ***U***, within which Object C moves at the constant velocity ***U***${}_{C}$, in both cases measured in Cartesian coordinates with respect to the reference solid. The macroscopic entropy production (Equation 10) is now given by the two- or three-dimensional vector scalar product:
(42)$$\begin{array}{c} {\bar{\dot{\sigma }}}_{ext}=\frac{F_{C}\cdot \left(U-U_{C}\right)}{T} \end{array}$$
where $F_{C}$ is the drag-lift force vector, containing drag and lift components aligned with and normal to the direction of motion, respectively (in three-dimensional systems there can be two lift components). Equation (Equation 42) represents the combined effect of frictional and pressure forces. The parameters in (Equation 42) can be expressed in terms of a vector Reynolds number and vector drag-lift coefficient, to give:
(43)ReC=ρdC(U−UC)μ,CDC=FC12ρAC||U−UC||2,Σ˙¯C=12DC·ReC||ReC||2GaC
where $\left||X|\right|=\sqrt{X\cdot X}$ is the magnitude of vector X. As evident from (Equation 42) and (Equation 43), there must be an acute angle π≤θ≤π between vectors $F_{C}$ and $U-U_{C}$ (or between $C_{D_{C}}$ and ReC), to ensure the nonnegativity of the entropy production. From the Kutta-Joukowski theorem, in a potential flow the lift component of $F_{C}$ will be proportional to the fluid circulation $\Gamma =-\oint_{C}u\cdot ds$ on any closed path C around the object, where s is the intrinsic coordinate [47,48,52,53,62]. Since a two-dimensional asymmetric object (e.g., an airfoil or hydrofoil) creates a non-zero fluid circulation, it will produce lift. For an object with no lift forces, $F_{C}$ and $U-U_{C}$ will be oriented in the same direction, and we recover ${\bar{\dot{\Sigma }}}_{C}=\frac{1}{2}C_{D_{C}}{Re_{C}}^{3}/Ga_{C}$ in (Equation 36) based on scalar variables.Using a mechanism mounted on Object C (e.g., a sail), the ambient flow can be harnessed to extract the negentropy required for motion in almost any direction, but only to some maximum extent ${\bar{\dot{\Sigma }}}_{C^{\prime }}$ that is harnessable from the flow (allowing for the possibility of direction-dependent parameters such as areas, length scales, and drag coefficients). Beyond this, any positive excess ${\bar{\dot{\Sigma }}}_{C}-{\bar{\dot{\Sigma }}}_{C^{\prime }}$ must have an independent source of negentropy, most likely attached to the object. Similar arguments apply to other external flow subsystems. Reexamining Object B, we note it may be very difficult to harness the ambient flow to enable countercurrent motion, but it is certainly possible to facilitate motion on an oblique reverse trajectory and to thereby construct a zigzag course (“tacking“) to achieve a net upstream motion. In all cases, the negentropy not actually harnessed from the ambient flow or extracted by a connection to another solid must be provided by the independent source of negentropy associated with the object.(d)*Subsystem D*: Now consider Subsystem D, an external flow subsystem within an internal flow subsystem, in turn situated within the main flow field. This preferences the entropic pair of inertial frames defined by its internal flow and Object D, with the inertial separation $U_{D}$. For the example shown in Figure 3, the surrounding ambient flow field does not directly affect the fluid inertial frame of reference, but due to the connection between Object D and the reference solid, the subsystem has the same solid inertial frame of reference as the ambient flow. For any nonzero $U_{D}$, the subsystem will have a positive $Re_{D}$, $C_{D_{D}}$ and entropy production ${\bar{\dot{\Sigma }}}_{D}$ due to the drag force on Object D, and also a positive $Re_{\vec{D}}$, $f_{\vec{D}}$, and entropy production ${\bar{\dot{\Sigma }}}_{\vec{D}}$ due to the internal flow within Subsystem D. In dimensional form, from (Equation 32) the total is:
(44)$$\begin{array}{c} {\bar{\dot{\sigma }}}_{D}+{\bar{\dot{\sigma }}}_{\vec{D}}=\frac{A_{D}\mu g}{T_{D}}\;{\bar{\dot{\Sigma }}}_{D}+\frac{d_{\vec{D}}L_{\vec{D}}\mu g}{T_{\vec{D}}}\;{\bar{\dot{\Sigma }}}_{\vec{D}} \end{array}$$Due to the physical connection between Object D and the reference solid, we know that the source of negentropy for (Equation 44) cannot reside with Object D. It is possible that negentropy could be harnessed from the bulk flow, but only to the maximum extent that is harnessable, for example from (Equation 10):
(45)$$|{\bar{\dot{\varphi }}}_{{\vec{D}}^{\prime }}|={\bar{\dot{\sigma }}}_{{\vec{D}}^{\prime }}=\frac{P_{{\vec{D}}^{\prime }}}{T_{{\vec{D}}^{\prime }}}$$
where $P_{{\vec{D}}^{\prime }}$ is the maximum power extractable by Subsystem $\vec{D}$ and $T_{{\vec{D}}^{\prime }}$ is its reference temperature. For example, considering only the kinetic energy component of the ambient flow, for Subsystem D with the intake area $A_{\vec{D},in}$, by continuity $Q_{\vec{D}}=A_{\vec{D},in}U$, so from (Equation 30), ${\bar{\dot{\sigma }}}_{{\vec{D}}^{\prime }}=\rho gH_{L,\vec{D}}Q_{\vec{D}}/T_{{\vec{D}}^{\prime }}=\rho gH_{L,\vec{D}}A_{\vec{D},in}U/T_{{\vec{D}}^{\prime }}$. In general, the form of (Equation 45) will depend on the design of Subsystem D and its conversion efficiency, so this is not analyzed further here. Here, it is sufficient to state that the excess $\left({\bar{\dot{\sigma }}}_{D}+{\bar{\dot{\sigma }}}_{\vec{D}}\right)-{\bar{\dot{\sigma }}}_{{\vec{D}}^{\prime }}$ between (Equation 45) and (Equation 44), if positive, must be provided by an independent source of negentropy, which in this example must power the mechanism (such as a pump, blower, or compressor) used to create the flow field for Subsystem D. If no negentropy is harnessed from the bulk flow, then all of the total ${\bar{\dot{\sigma }}}_{D}+{\bar{\dot{\sigma }}}_{\vec{D}}$ must be provided by this independent source of negentropy.(e)*Other Variants*: If we now release Object D from its solid connection, we create an independent external flow subsystem within the internal flow of Subsystem D. At steady state, Object D could harness negentropy from the Subsystem D flow field to provide for its entropy production ${\bar{\dot{\sigma }}}_{D}$ but only to the extent ${\bar{\dot{\sigma }}}_{D^{\prime }}$ that this is harnessable. Any excess ${\bar{\dot{\sigma }}}_{D}-{\bar{\dot{\sigma }}}_{D^{\prime }}$ must be incurred by yet another independent source of negentropy attached to Object D, to maintain it in a fixed position.Finally, we can imagine Subsystem D to be in steady motion with respect to the bulk flow field, thus involving several nested external and internal flow systems. This will require sources of negentropy for the entropy production of all component subsystems. By connections or coupling, some of these could extract negentropy from their surrounding systems, but only to the extent possible, with any excess requiring one or more independent sources of negentropy.

#### 4.3.3. Governing Principles

From the above examples, we can draw out the following governing principles:
(1)(A)*A shear flow subsystem at steady state, defined by fluid flow relative to a solid object, preferentially selects an* entropic pair *of inertial frames of reference, consisting of (i) the frame moving with the solid, i.e., that in which the solid is stationary, and (ii) the frame moving with the mean fluid flow, i.e., that in which the mean fluid flow is stationary. These frames are unique. They can be described as being* entropically paired *by the subsystem and* inertially separated *by their difference in velocities.*
(2)(A)*An entropically paired shear flow subsystem must have at least one source of negentropy for its entropy production.*
(3)*If the entropic pair of a shear flow subsystem differs from the frames of reference that define the surrounding flow, then:*(A)*The shear flow subsystem may be harnessing negentropy from its external environment, either from the ambient flow field or by exploiting some other connection, but it can only do so to the extent that these are harnessable by the subsystem.*(B)*Any excess entropy production, above what is actually harnessed from the external environment, reveals the existence of at least one independent source of negentropy for the shear flow subsystem.*

We further note that the solid and fluid flow frames of reference that define an entropic pair *need not be contained within the subsystem*, but can be nonlocal to that subsystem. Indeed, for the external flow systems shown in Figure 1 and Subsystems A to C in Figure 3, the ambient flow field is referenced at a long distance upstream. Furthermore, in all subsystems shown in Figure 1, Figure 2 and Figure 3, the no-slip condition will require the fluid velocity to vanish at the solid surface, so there must be a physical separation between the solid and fluid flow which define the entropic pair. In all cases described, the nonlocal effects are transmitted by the flow field.

The above examples become more more complicated if we also include the effects of gravitational, buoyancy, or lift forces, and the requirements of thermodynamic efficiency. For objects lighter or denser than the fluid or which experience a lift force, maintaining its vertical position will make an additional contribution to the entropy production. This will add to the negentropy required by the subsystem (see Section 5). Similarly, for processes of less than 100% efficiency, there must be a source of negentropy for the lost work component, which must be taken into account in the above calculations. This can be represented by a modified rate of negentropy consumption for each process, in dimensional or time-averaged dimensionless form:(46)$$\begin{array}{c} |\dot{\varphi }|=\frac{\dot{\sigma }}{\eta }\geq 0\;\mathrm{or}\;\;\left|\bar{\dot{\Phi }}\right|=\frac{\bar{\dot{\Sigma }}}{\eta }\geq 0 \end{array}$$
where 0≤η≤1 is the thermodynamic efficiency. Additional complications will arise in flow fields with nonuniform fluid or thermodynamic parameters, such as velocity, temperature, fluid density, or dynamic viscosity, and in subsystems with acceleration.

The phenomena revealed in this section are very different to the motions of frictionless objects in Newtonian mechanics, which do not require the consumption of a source of negentropy for an object or fluid to maintain a constant velocity. Taken together, they provide new perspectives into the long-neglected field of *entropic mechanics*, the science of the relative motions of objects with friction. As hinted at in the above discussion, this will necessarily include the motions of all motor vehicles and other craft (irrespective of power source) and of all living organisms, relative to any fluid.

### 4.4. Other Steady-State Shear Flow Systems

The foregoing arguments apply with some modification to all other shear flow systems at steady state. Consider the following idealized classes of shear flow systems, as illustrated in Figure 4 [5,46,48,49,51,52,88,89]:(a)In a *two-dimensional or three-dimensional external flow*, as shown in Figure 1 or Figure 4a, the entropic pair consists of the solid and the ambient flow, the latter referenced to its upstream mean velocity profile. In the *turbulent wake* downstream of the solid, the flow field is subject to nonlocal influences of both the original fluid stream and the solid. As discussed in Section 4.1.1, there must be at least one source of negentropy associated with the solid and/or the ambient flow, to maintain the steady-state flow.(b)In a *two-dimensional or three-dimensional internal flow* under a pressure gradient (*Poiseuille flow*), as shown in Figure 2 or Figure 4b, the entropic pair consists of the solid wall(s) and the internal flow, the latter referenced at its mean velocity. As discussed in Section 4.1.2, there must be at least one source of negentropy, associated with the solid wall(s) and/or the fluid flow, to maintain the steady-state flow.(c)In *Couette flow*, involving the relative motion of parallel plates or concentric cylinders in contact with a fluid as shown in Figure 4c, the entropic pair is provided by the two solid walls of the system. At least one of the solids must have an independent source of negentropy, such as an engine connected to a driveshaft or crankshaft, to drive the relative motion, which in turn generates the fluid flow.(d)In a combined *Couette-Poiseuille flow*, consisting of fluid flow under a pressure gradient between moving parallel plates or concentric cylinders, as shown in Figure 4d, the two solids and the internal flow (referenced at its mean velocity) provide an *entropic triple* of inertial frames of reference for the system. From the above principles, there must be at least two independent sources of negentropy: one for the internal flow, and at least one for the relative motions of the solids.(e)For a *boundary layer flow*, consisting of fluid flow relative to a solid boundary such as that as shown in Figure 4e, the flow is commonly analyzed using the Reynolds number $Re_{x}=\rho xU_{\infty }/\mu$ or $Re_{\delta }=\rho \delta U_{\infty }/\mu$, based on the distance *x* [m] or boundary-layer thickness δ [m] from the start of the plate, respectively [5,46,47,48,49,50,51,52]. These Reynolds numbers are functions of position, but all contain the same reference velocity $U_{\infty }$. The entropic pair is therefore provided by the solid and the ambient flow (referenced to the upstream flow field or at infinite vertical distance). The system must have at least one source of negentropy, which could be associated with the solid object and/or the ambient flow field.(f)Consider the *two-dimensional turbulent mixing layer*, in which two fluid streams separated by a solid plate are allowed to merge beyond the end of the plate, as shown in Figure 4f. Here, the two fluid streams (referenced at infinite distance) and solid provide an entropic triple for the system. This example illustrates the nonlocal influence of a solid on the downstream fluid—its influence cannot be neglected even for long distances downstream. There must be at least two independent sources of negentropy associated with this triple, to drive the two independent flows.(g)Consider the *two-dimensional or axisymmetric turbulent jet* issuing into a stationary fluid, as shown in Figure 4g. Here, the solid nozzle and its internal flow field (referenced at its mean velocity) provide the entropic pair for the system. This system again illustrates the nonlocal influence of a solid to create the wedge-shaped or conical zone of fluid flow produced by shear against the ambient fluid. The fluid flow must be driven by at least one independent source of negentropy, associated with the fluid jet and/or solid nozzle.(h)Consider the *two-dimensional or axisymmetric turbulent jet* issuing into an ambient parallel flow, as shown in Figure 4h. Here the solid nozzle and two fluid flows provide an entropic triple for the system. There must be at least two independent sources of negentropy associated with this triple, to drive the two fluid flows.

From these examples, we can add the following governing principles to those listed in Section 4.3.3:
(1)(B)*A shear flow subsystem at steady state, generated by the relative motion of two solid objects within a fluid, preferentially selects an* entropic pair *of inertial frames of reference, consisting of the frames moving with each solid.*(C)*A shear flow subsystem at steady state, defined by the relative motions of two solid objects and a fluid stream, or two fluid streams and a solid object, preferentially selects an* entropic triple *of inertial frames of reference, consisting of the frames moving with each solid and fluid flow. The entropic triple can be analyzed in terms of its three constituent entropic pairs.*(2)(B)*An entropically tripled shear flow subsystem must have at least two independent sources of negentropy for its entropy production.*
We can also modify principle (3) to include entropically tripled systems. Similar considerations apply to more complicated fluid steady-state flow systems, for example, the jet boundary, the double boundary layer, the turbulent jet issuing into a cross-flow, or the buoyant or dense jet, c.f. [5,46,48,49,51,52,88,89].

The above analysis can be extended *ad infinitum*. Consider the *multiple mixing layer* consisting of *m* independent flows separated by *n* parallel plates, as shown in Figure 5a. Assuming that the flows are generated by independent pressure gradients, and that the solids are not connected, this system preferentially selects the *entropic (m+n)-tuple* of inertial frames of reference moving with each fluid flow and each solid. This must have at least (m+n−1) independent sources of negentropy, to drive the relative fluid and solid motions. If, however, there are dependencies or connections between the fluid flows or solids, such as for flow through a set of nested cylinders or a lattice, the number of degrees of freedom of the entropic tuple and the number of independent sources of negentropy will diminish accordingly.

This example provides the additional governing principles:
(1)(D)*A shear flow subsystem at steady state, defined by the relative motions of m independent solid objects and n independent fluid streams, preferentially selects an* entropic *(m+n)*-tuple *of inertial frames of reference, consisting of the frames moving with each solid and fluid flow. The entropic (m+n)*-tuple *can be analyzed in terms of its $\frac{1}{2}\left(n+m\right)\left(n+m-1\right)$ constituent entropic pairs.*(2)(C)*An entropically (m+n)-tupled shear flow subsystem must have at least (m+n−1) independent sources of negentropy for its entropy production.*
We can also modify principle (3) to include entropically tupled systems of any order.

Finally, consider a small shear flow subsystem at steady state within a complicated but steady flow field, as shown in Figure 5b. Here, the ambient flow has been influenced by *k* upstream solids or fluid flows, hence, the external flow subsystem preferentially selects an entropic (k+1)-tuple of inertial frames of reference. Nonetheless, the governing principles 2(A) and (3)(A)–(B) in Section 4.3.3 still apply; if the entropic pair of the subsystem differs from the inertial frames of reference that define the ambient flow, the subsystem must have an independent source of negentropy for its excess entropy production, above what is harnessed from the ambient flow. In the example shown, if the solid object is connected to the reference solid, its entropy production will be borne by the ambient flow field; however, if not, then it must have its own independent source of negentropy to maintain its stationary position or motion, in accordance with the analyses in Section 4.3.2. Similar arguments apply to systems with higher order influences, such as an entropically *ℓ*-tupled shear flow subsystem embedded within an entropically (k+1)-tupled flow system.

## 5. Unsteady Shear Flow Systems

We now consider unsteady shear flow systems or, in other words, systems with acceleration, including changes in speed and/or direction. This necessarily includes all shear flows with rotation. For the sake of brevity, we restrict the discussion to unsteady extensions of the flows examined in Section 4.1. For these systems, it is appropriate to include the action of body forces due to a gravitational or electromagnetic field, omitted from the idealized steady-state flows in Section 4.

### 5.1. Unsteady External Flows

Consider the two- or three-dimensional *unsteady external flow* represented by Figure 6a. This encompasses a wide variety of flow systems with different reference frames, from a fluid in motion relative to a stationary object (e.g., flow past a model in a wind tunnel), through to a solid object in motion relative to a stationary fluid (e.g., a soccer ball in flight).

First consider the unsteady purely translational (irrotational) motion of an isolated rigid sphere with velocity V(t) [m s${}^{-1}$] in a uniform flow field of velocity U(t) [m s${}^{-1}$], both relative to a common inertial frame of reference (Figure 6a without rotation). A force balance at low velocities yields [90,91,92,93,94,95,96,97,98]:(47)$$\begin{array}{c} m_{s}\frac{dV}{dt}=F_{P}+F_{B}+F_{D}+F_{I}+F_{H}+F_{F} \end{array}$$
where $m_{s}$ [kg] is the mass of the solid, $F_{P}$ [N] is a propulsion force on the object, $F_{B}$ [N] is the net body force (e.g., gravity minus buoyancy), $F_{D}$ [N] is the drag-lift force on the object due to viscous and pressure forces, $F_{I}$ [N] is the inertial force due to the “added mass” of fluid accelerated by the object, $F_{H}$ [N] is a history-dependent force to account for acceleration memory effects, and $F_{F}$ is the fluid force due to acceleration of the local fluid. The total resistance force on the solid is $F_{D}+F_{I}+F_{H}+F_{F}$, hence, the entropy production is:(48)$$\begin{array}{cc} {\dot{\sigma }}_{ext}^{trans}\;\left(t\right)\; & =\frac{\left(F_{D}+F_{I}+F_{H}+F_{F}\right)\cdot \left(U-V\right)}{T} \end{array}$$
assuming isothermal conditions. Equation (Equation 48) can be nondimensionalized in a manner similar to steady-state flows (Equation 43) to give the dot product between the vector Reynolds number Res=ρfds(U−V)/μ and a sum of vector drag-lift coefficients, each moderated by a function of the norm of its corresponding velocity or acceleration term.

Now consider purely rotational motion, in which a rigid sphere of radius vector r [m] rotates about its centroid at the angular velocity ω(t) [s${}^{-1}$], due to a torque T [N m] on the solid (Figure 6a without translation). The entropy production is:(49)$$\begin{array}{c} {\dot{\sigma }}_{ext}^{rot}\left(t\right)=\frac{T\cdot \omega }{T} \end{array}$$
assuming isothermal conditions. Equation (Equation 49) can be nondimensionalized as the dot product between a vector rotational Reynolds number (Taylor number) ReT=ρf||r||2ω/μ and a vector torque coefficient, c.f., [99,100], moderated by the norm of an angular acceleration term. For different solid shapes or centers of rotation, more comprehensive torque equations can be derived based on moments of inertia.

Now consider the combined unsteady translational and rotational system, as shown in Figure 6a. The entropy production is given by a combination of (Equation 48) and (Equation 49), with additional terms to account for coupling effects such as rotation-induced lift (the Magnus force). In Section 4.3.2(c), we saw that an object with mirror asymmetry (e.g., an airfoil) will create a non-zero fluid circulation, thereby inducing lift. Extending this idea, an object with rotational symmetry and mirror asymmetry, mounted on a fixed axis (e.g., a turbine), will interact with the flow to produce a torque on the object, thereby harnessing negentropy from the flow to cause its rotation. Even without the fixed mounting, such an object will undergo rotation as well as translation, thus harnessing the flow for a proportion of its entropy production. For complex solids and/or nonuniform flow fields, the drag-lift and torque coefficients are generally found by numerical integration of pressures and viscous stresses around the solid surface, calculated using a turbulence model [101,102]. For flow-induced vibrations, the coefficients will also be functions of the Strouhal number $St=fd_{s}/\left|\right|U-V||$, where f is the vibration frequency [s^−1^] [50,52,103,104]. Considerable research is now underway on more complicated systems such as tethered solids with various degrees of freedom, of interest to the study of fluid–structure interactions, e.g., [105].

Finally, transonic and supersonic flows (travelling close to or faster than the speed of sound) cause the formation of shock waves, with sudden changes in velocity, pressure, density, and temperature [47,52,62]. With an increasing Mach number, these cause a sharp increase in drag, modify the lift, and cause significant heating, substantially increasing the entropy production.

In these unsteady external flows, the system will preferentially select the frames of reference moving with the reference fluid flow and solid at each time instant. For translational motion, this defines a single entropic pair made up of the flow field and solid, while for rotational motion, it defines a joint or disjoint continuous set of entropic pairs, consisting of the flow field and each point on the solid surface. However, the flow field and/or the solid within each entropic pair will change with time due to the unsteady motion. In addition, at least one of the frames from each entropic pair, and possibly both, will be a non-inertial frame of reference. In consequence, we can add the following governing principle to those given previously:
(1)(E)*A shear flow subsystem in unsteady flow, defined by the relative motions of a fluid and a solid, preferentially selects an entropic pair of frames of reference—or a set of entropic pairs—that is changing in time. At least one frame from each entropic pair will be a non-inertial frame of reference.*
From principles 2(A) and 3(A)–(B) from Section 4.3.3, the system must have an independent source of negentropy for its entropy production, above what is being harnessed from the flow. Given the possibility of inducing rotation of an object to harness negentropy from the flow, which can then be used to power translational motion, it is not possible to be too definitive as to the location of these source(s) of negentropy. We can however make the following addition to governing principles 3(A)–(B): (3)*If the entropic pair (or other entropic tuple) of a shear flow subsystem differs from the frames of reference that define the surrounding flow, including changes with time, then:*(A)*The shear flow subsystem may be harnessing negentropy from its external environment, either from the ambient flow field or by exploiting some other connection, but can only do so to the extent that these are harnessable by the subsystem.*(B)*Any excess entropy production, above what is actually harnessed from the external environment, reveals the existence of at least one independent source of negentropy for the shear flow subsystem.*

### 5.2. Unsteady Internal Flows

Now consider the two- or three-dimensional *unsteady internal flow* represented by Figure 6b, in which a fluid moves at the instantaneous velocity u(x,t) and mean velocity U(t) relative to the solid walls, which are stationary with respect to an inertial frame of reference. This can again be represented by the scalar Reynolds number (Equation 28), pressure loss (Equation 29), and entropy production (Equation 30) and (Equation 31) for internal flows, now as functions of time. Such flows can be divided into two classes. For a rigid fluid and solid walls, slow changes in velocity, such as the onset of flow, can be calculated from the one-dimensional momentum equation [48,49,52]. In contrast, for an elastic fluid and solid walls, a sudden change in velocity will create pressure waves migrating in both directions, for which the wave speed can be determined from the continuity and momentum equations [48,49,52]. Both flows will alter the instantaneous entropy production (Equation 30) and (Equation 31), predominantly through the cubic velocity term. In transonic and supersonic internal flows, the formation of shock waves will also substantially increase the entropy production as a function of the Mach number.

A second category of internal flows is provided by the open channel flow of a liquid under gravity, bounded by solid walls and a free liquid surface [48,49,52]. Uniform steady channel flows can be analyzed by channel variants of the Reynolds number (Equation 28), pressure loss (Equation 29), and entropy production (Equation 30) and (Equation 31), expressed in terms of the water surface elevation rather than pressure head, requiring solution of the continuity and momentum equations [106,107,108]. Nonuniform or unsteady channel flows require successive additional terms in the momentum equation [106,107,108], while oscillatory surface waves require a different treatment beyond the scope of the current discussion [107,108]. Channel flows with obstacles have the features of both external and internal flows, requiring the synthesis of (Equation 26) and (Equation 30). As with unsteady external flows in Section 5.1, a rotationally symmetric object mounted on a fixed axis, intruding into a channel (e.g., a water wheel), can harness negentropy from the flow in the form of rotational motion. Similarly, an object in contact with a solid wall (e.g., a sediment particle) can harness the flow for its transport, by suspension, a bouncing motion (saltation), tumbling, rolling, or sliding [107,109]. Finally, critical and supercritical channel flows (travelling at or faster than the speed of a surface wave) can cause the formation of an hydraulic jump, with a sudden increase in water level, substantially increasing the entropy production as a function of the Froude number.

The above unsteady internal flows exhibit similar properties to the external flows of Section 5.1 in that they preferentially select an entropic pair of frames of reference (or a set of pairs) that change with time. In consequence, we can draw the same conclusions as those previously, concerning the selection of entropic pair(s) and the source(s) of negentropy for each subsystem.

## 6. Conclusions

This study examined the invariance properties of the thermodynamic entropy production, based on its global (integral), local (differential), local bilinear, and global macroscopic formulations, as defined in Section 2. The mathematical invariance properties of these equations were examined in Section 3, including dimensional scaling, invariance to fixed coordinate displacements, rotations or reflections, Galilean invariance, and the one-parameter Lie group of point transformations. Of these, the Lie invariance can be reinterpreted as a generalized dimensionless invariance, which reveals and is expressed in terms of the intrinsic or ‘indigenous’ dimensions of the system. The Lie invariance can also be shown to encompass the other invariances, and is therefore the most general.

We then examined a number of shear flow systems involving relative motion between fluid(s) and solid(s), first for steady-state flow (Section 4) and then for unsteady flow (Section 5). In a steady-state shear flow system consisting of a single fluid and solid, the Galilean invariance property was shown to preference a unique pair of inertial frames of reference for the system—here referred to as an *entropic pair*—from the infinite set of available reference frames for the system. This challenges the Newtonian viewpoint that all inertial frames of reference are equivalent. This entropic pairing can be considered to be an additional invariant property of the system, enabling the Reynolds number, drag coefficient, and entropy production to be uniquely defined and Galilean invariant. In Section 4.3, we drew out an apparent paradox arising from the coexistence of different entropic pairs for different shear flow subsystems within a flow system. For each subsystem, the paradox was resolved by the fact that it reveals the presence of at least one independent source of negentropy—a power source scaled by the absolute temperature—being depleted by one or more of the different parts. By the analysis of a variety of steady-state and unsteady shear flow subsystems, we drew out a series of governing principles to describe their entropic pairing properties and sources of negentropy. These are reiterated here in consolidated form:
(1)(A)*A shear flow subsystem at steady state, defined by fluid flow relative to a solid object, preferentially selects an* entropic pair *of inertial frames of reference, consisting of (i) the frame moving with the solid, i.e., that in which the solid is stationary, and (ii) the frame moving with the mean fluid flow, i.e., that in which the mean fluid flow is stationary. These frames are unique. They can be described as being* entropically paired *by the subsystem and* inertially separated *by their difference in velocities.*(B)*A shear flow subsystem at steady state, generated by the relative motion of two solid objects within a fluid, preferentially selects an* entropic pair *of inertial frames of reference, consisting of the frames moving with each solid.*(C)*A shear flow subsystem at steady state, defined by the relative motions of two solid objects and a fluid stream, or two fluid streams and a solid object, preferentially selects an* entropic triple *of inertial frames of reference, consisting of the frames moving with each solid and fluid flow. The entropic triple can be analyzed in terms of its three constituent entropic pairs.*(D)*A shear flow subsystem at steady state, defined by the relative motions of m independent solid objects and n independent fluid streams, preferentially selects an* entropic *(m+n)*-tuple *of inertial frames of reference, consisting of the frames moving with each solid and fluid flow. The entropic (m+n)*-tuple *can be analyzed in terms of its $\frac{1}{2}\left(n+m\right)\left(n+m-1\right)$ constituent entropic pairs.*(E)*A shear flow subsystem in unsteady flow, defined by the relative motions of a fluid and a solid, preferentially selects an entropic pair of frames of reference—or a set of entropic pairs—that is changing in time. At least one frame from each entropic pair will be a non-inertial frame of reference.*(2)(A)*An entropically paired shear flow subsystem must have at least one source of negentropy for its entropy production.*(B)*An entropically tripled shear flow subsystem must have at least two independent sources of negentropy for its entropy production.*(C)*An entropically (m+n)-tupled shear flow subsystem must have at least (m+n−1) independent sources of negentropy for its entropy production.*(3)*If the entropic pair (or other entropic tuple) of a shear flow subsystem differs from the frames of reference that define the surrounding flow, including changes with time, then:*(A)*The shear flow subsystem may be harnessing negentropy from its external environment, either from the ambient flow field or by exploiting some other connection, but can only do so to the extent that these are harnessable by the subsystem.*(B)*Any excess entropy production, above what is actually harnessed from the external environment, reveals the existence of at least one independent source of negentropy for the shear flow subsystem.*
The above principles are unaffected by Galilean transformations and so can be understood to “lie above” the Galilean inertial framework of Newtonian mechanics.

The phenomena revealed in this study are very different to the motions of frictionless objects in Newtonian mechanics, which do not require the consumption of a source of negentropy for an object or fluid to maintain a constant velocity. Taken together, they provide new perspectives into the long-neglected field of *entropic mechanics*, the study of the relative motions of objects with friction. This encompasses the motions of all motor vehicles and other craft (irrespective of power source) and of all living organisms within a fluid, whether they be in the atmosphere, on the land surface, or at any location on the surface of, within, or at the base of a water body or any other liquid.

Further research is required to elucidate the complete set of Lie symmetries associated with the entropy production equations, including multivariate Lie symmetries associated with multivariate continuous groups [26,58,59]. Further research is also warranted on analogs of the entropic pairing principles for other types of dissipative systems, including heat transfer, chemical reaction, and living systems [34,35,38,110], and their implications for the sources of negentropy that drive these systems.

## Figures and Tables

**Figure 1 entropy-23-01515-f001:**
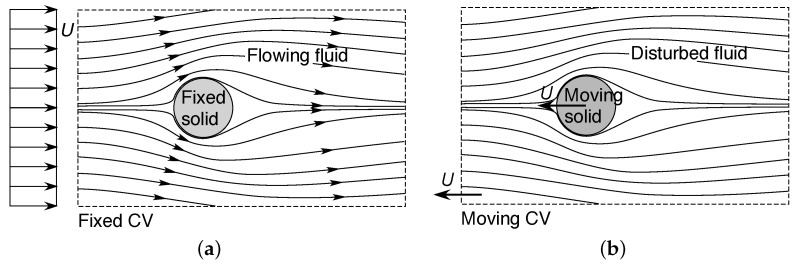
Cross-sections of two representations of an external flow system at steady state, for the inertial frame of reference (**a**) relative to the solid or (**b**) relative to the fluid (compare [61]). In both cases, the fluid extends beyond the rectangular control volume shown.

**Figure 2 entropy-23-01515-f002:**
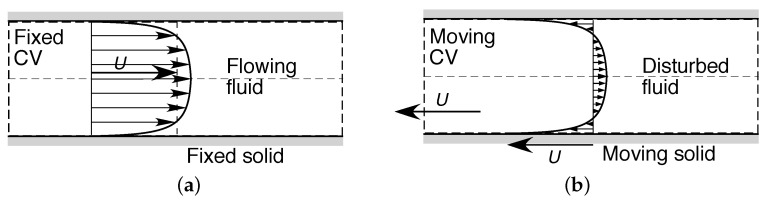
Two representations of an internal flow system at steady state, in the region of developed flow, for the inertial frame of reference (**a**) relative to the solid or (**b**) relative to the fluid. The velocity arrows are drawn relative to the chosen inertial frame of reference.

**Figure 3 entropy-23-01515-f003:**
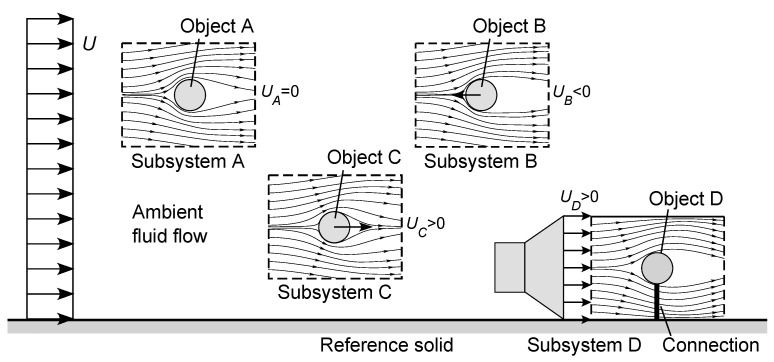
Multiple external flow subsystems within a flow system adopting an inertial frame of reference relative to a reference solid (compare [61]). For the subsystem control volumes, permeable boundaries are drawn with dashed lines, and impermeable boundaries with solid lines. All velocities are defined relative to the reference solid.

**Figure 4 entropy-23-01515-f004:**
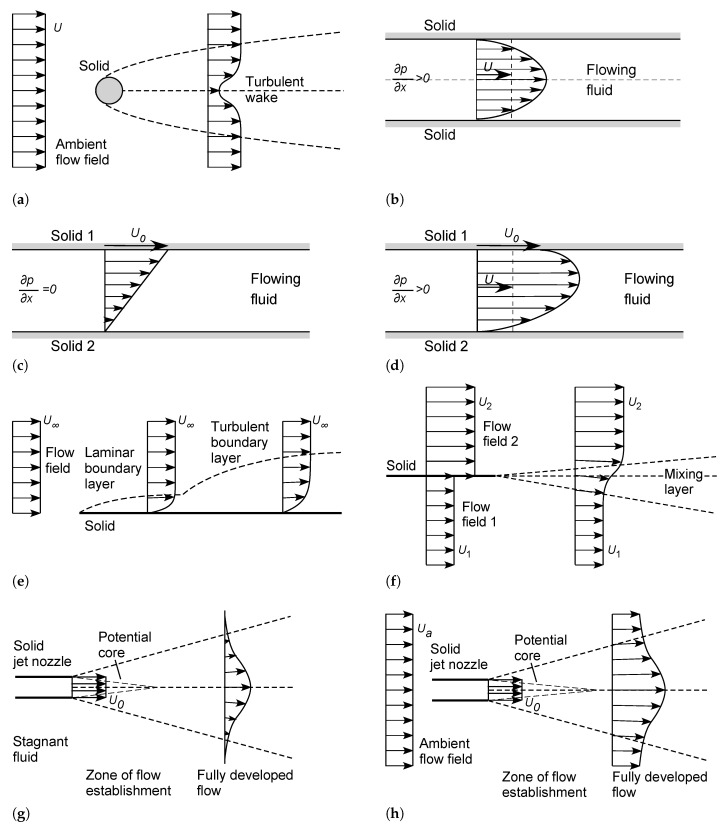
Schematic diagrams of several idealized classes of shear flow systems: (**a**) external flow and the turbulent wake (c.f., Figure 1), (**b**) internal (Poiseuille) flow (c.f., Figure 2), (**c**) Couette flow, (**d**) combined Couette-Poiseuille flow, (**e**) boundary layer flow, (**f**) the two-dimensional mixing layer, (**g**) the two-dimensional or axisymmetric turbulent jet, and (**f**) the two-dimensional or axisymmetric turbulent jet issuing into a parallel ambient flow, e.g., [5,46,48,49,50,51,52,88,89]. The time-averaged velocity profiles are all drawn to include the small transverse or radial velocity component, if present.

**Figure 5 entropy-23-01515-f005:**
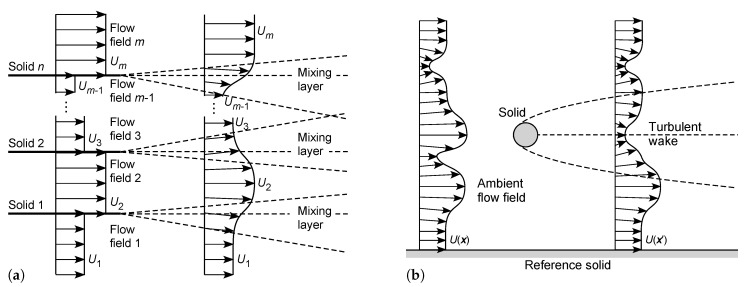
Schematic diagrams of more complicated shear flow systems: (**a**) the multiple mixing layer, and (**b**) a generalized shear flow subsystem.

**Figure 6 entropy-23-01515-f006:**
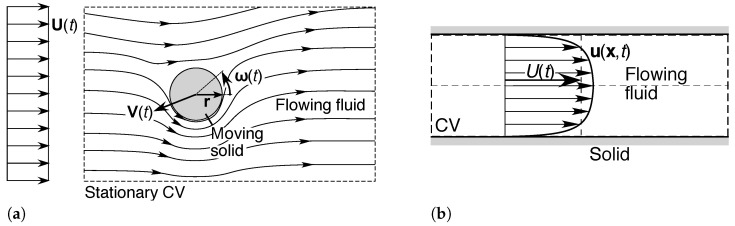
Schematic diagrams of unsteady shear flow systems: (**a**) unsteady external flow and (**b**) unsteady internal flow. The velocities are expressed relative to an inertial frame of reference in which the control volume is stationary.

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
