# Peer review of "Invariance Properties of the Entropy Production, and the Entropic Pairing of Inertial Frames of Reference by Shear-Flow Systems"

_entropy, 2021, doi:10.3390/e23111515_

Round 1
Reviewer 1 Report
The main idea of the paper [ copied from the paper below] has already been published in Cemal Basaran, Introduction to Unified Mechanics Theory with Applications, 2021, Springer Nature Switzerland AG.
What is new in their paper it is not clear. They need to compare their work to the existing work in the cited book explains in the paper what they have new as a contribtion
"The above principles are unaffected by Galilean transformations, and so can be understood to “lie above” the Galilean inertial framework of Newtonian mechanics." " The phenomena revealed in this study are very different to the motions of frictionless objects in Newtonian mechanics, which do not require the consumption of a source of negentropy for an object or fluid to maintain a constant velocity. Taken together, they provide new perspectives into the long-neglected field of entropic mechanics, the study of the relative motions of objects with friction...."
Reviewer 2 Report
Dear Authors,
Please see the attached file.
I thank for your answers.
Best regards

Round 2
Reviewer 1 Report
Author answered all my questions